# Napthoquinones from *Neocosmospora* sp.—Antibiotic Activity against *Acidovorax citrulli*, the Causative Agent of Bacterial Fruit Blotch in Watermelon and Melon

**DOI:** 10.3390/jof7050370

**Published:** 2021-05-08

**Authors:** Anthikan Klomchit, Jorge Daniel Calderin, Wuttichai Jaidee, Kanchana Watla-iad, Siraprapa Brooks

**Affiliations:** 1School of Science, Mae Fah Luang University, Chiang Rai 57100, Thailand; 6151105007@lamduan.mfu.ac.th (A.K.); kanchana.wat@mfu.ac.th (K.W.-i.); 2Department of Biochemistry, University of Illinois, Urbana-Champaign, IL 61820, USA; Jorge.calderin003@gmail.com; 3Medicinal Plant Innovation Center, Mae Fah Luang University, Chiang Rai 57100, Thailand; wuttichai.jai@mfu.ac.th

**Keywords:** *Acidovorax citrulli*, endophytic fungi, antibacterial activity, secondary metabolites

## Abstract

Bacterial fruit blotch (BFB) is a bacterial disease that devastates *Cucurbitaceae* crops worldwide, causing significant economic losses. Currently, there is no means to treat or control the disease. This study focused on exploring the antibacterial properties of endophytic fungi against *Acidovorax citrulli* (*Aac*), the causative agent of BFB. Based on disc diffusion, time kill and MIC microdilution broth assays, four endophytes showed promise in controlling *Aac*. Nonetheless, only one strain, *Neocosmospora* sp. MFLUCC 17-0253, reduced the severity of disease on watermelon and melon seedlings up to 80%. Structure analysis revealed production of several compounds by the fungus. Three of these secondary metabolites, including mixture of 2-methoxy-6-methyl-7-acetonyl-8-hydroxy-1,4-maphthalenedione and 5,8-dihydroxy-7-acetonyl-1,4-naphthalenedione, anhydrojavanicin, and fusarnaphthoquinones B exhibited antagonistic activity against *Aac*. The chemical profile data *in planta* experiment analyzed by LC-Q/TOF-MS suggested successful colonization of endophytic fungi in their host plant and different metabolic profiles between treated and untreated seedling. Biofilm assay also demonstrated that secondary metabolites of *Neocosmospora* sp. MFLUCC 17-0253 significantly inhibited biofilm development of *Aac*. To the best of our knowledge, secondary metabolites that provide significant growth inhibition of *Aac* are reported for the first time. Thus, *Neocosmospora* sp. MFLUCC 17-0253 possesses high potential as a biocontrol agent for BFB disease.

## 1. Introduction

Bacterial fruit blotch (BFB) is a serious seed borne disease that is destructive to cucurbit crops worldwide, especially watermelon (*Citrullus lanatus*) and melon (*Cucumis melo*). Seeds contaminated with the Gram-negative bacterium *Acidovorax citrulli*, formerly known as *Pseudomonas pseudoalcaligenes* subsp. *citrulli* [1] spread the disease. Infected areas appear to be greased, soaked, stained with an olive-green tint on the cotyledon and mature foliage, or blotched on the upper surface of the fruit, which makes it unmarketable [2,3]. The first outbreak of BFB occurred in 1989 in watermelon plantations in the United States, resulting in a 90% crop loss [4,5]. The primary source of BFB outbreaks is contaminated seeds, which causes significant economic loss for seedling production. The loss is especially pronounced in environments that are favorable for BFB infection, such as those of high humidity and temperature [6,7]. Chemical and biochemical approaches have been used to decontaminate cucurbit seeds, and attempts have been made to develop resistant varieties [8]. Unfortunately, there are no commercial cucurbit cultivars that are resistant to this disease [6,9]. Previous studies attempted to eliminate seed borne BFB by using oxalic acid produced from *Aspergillus niger* strain Y-1 to disinfect infected seeds. However, this method was of limited value as some bacteria escaped from within the seed coat, which negatively impacted seed germination and seedling growth [7,8,10]. Thus, currently, there are no known effective measures to manage BFB of commercially cultivated crops.

Endophytes are microorganisms that live within plant tissues for their entire life cycle without causing infection or signs of disease [11]. Fungal endophytes are known for their capability in inhibiting plant pathogens including destructive bacteria, fungi, viruses, and insects [12,13]. Several mechanisms including antibiotic production, induction of systemic resistance, and competition for nutrients and space are some of the biological functions of endophytes [14,15]. One of the most notable attributes of endophytes is their ability to produce bioactive secondary metabolites and enzymes, which are considered valuable sources in pharmaceutical industry and agriculture [16,17,18]. For instance, several active naphthoquinone derivatives that were isolated from *Fusarium acutatum* had insecticidal, antibacterial, and fungicidal activities [19,20]. Silva et al. (2006) reported that the active compound 3,11,12-trihydroxycadalene, produced by endophytic fungi isolated from *Cassiae spectabillis*, could inhibit growth of the plant pathogens *Cladosporium sphaerospermum* and *C. cladosporioides* [21].

The composition and diversity of endophytic fungi within a plant is influenced by factors such as host species, geographic location, plant physiological status, season, and tissue/organ of the host plants [22,23,24]. Mangrove forests have adapted to extreme environmental conditions and harbor high biodiversity. Accordingly, mangrove hosts have a great variety of endophytes [25,26]. Mangrove endophytic fungi represent the second largest ecological group of marine-derived fungi, and are promising sources of bioactive natural products that can lead to the discovery of novel drugs [27]. For instance, cytosporone B and C were isolated from the mangrove endophytic fungus, *Phomopsis* sp. ZSU-H76. Both compounds exhibited antifungal activity and inhibited *Candida albicans* and *F. oxysporum* [28]. At the same time, several studies have focused on endophytic fungi associated with tea (*Camellia sinensis*), leading to the discovery of an array of bioactive natural products. For instance, over 70 new bioactive natural products were discovered from *Pestalotiopsis* spp., an endophytic fungus of *C. sinensis* [29]. Additionally, Guo et al. (2016) identified ten new butenolide derivatives from *Aspergillus terreus* isolated from *C. sinensis* var. *Assamica*, all of which had potent anti-inflammatory activity [30].

Given the economic impact of BFB disease, there is urgent need for reliable methods to control it or produce pathogen-free seeds. Thus, this study aimed to evaluate the inhibitory activities of fungal endophytes isolated from mangrove forests and tea plantations against BFB disease as well as to investigate secondary metabolites obtained from potential fungal endophytes. We hypothesized that secondary metabolites obtained from fungal endophytes would have ability to reduce BFB disease severity. Our study provides valuable information to develop alternative methods to control a disease that is considered a major threat to watermelon and melon worldwide.

## 2. Materials and Methods

### 2.1. The Aac Strain and Endophytic Fungi Used in this Study

The virulent strain *Acidovorax citrulli* (*Aac*), JT-0003 was kindly provided from Chia Tai Co., Ltd. (Bangkok, Thailand). The strain was cultured on nutrient agar (NA; Himedia, Mumbai, India) medium at 28 °C for 48 h. A single colony was picked and placed into nutrient broth (NB; Himedia, Mumbai, India) medium and then incubated at 28 °C under shaking condition (120 rpm) for 48 h. The *Aac* culture was suspended in NB and the OD 600 was measured until it reached 0.2 UFC/mL. Fifty strains of endophytic fungal cultures were received from Mae Fah Luang University Culture Collection (MFLUCC), Center of Excellence in Fungal Research, Thailand (Table 1).

### 2.2. Fermentation and Extraction of Endophytic Fungi

Fifty strains of endophytic fungi were sub-cultured in potato dextrose agar (PDA; Himedia, Mumbai, India) medium at 28 °C for seven days, and ten pieces (0.5 × 0.5 cm^2^) of mycelia agar plugs were inoculated in 500-mL Erlenmeyer flasks containing 300 mL of potato dextrose broth (PDB; Himedia, Mumbai, India) medium. Broth cultures were incubated at 28 °C under static conditions. After incubating for 3 weeks, culture broths were separated from mycelia using sterile sheet cloth. An equal volume of ethyl acetate (EtOAc) was added to broth cultures in a separatory funnel and the mixed solvent was allowed to stand for 20 min. After filtration, the EtOAc was removed using a rotary evaporator under reduced pressure at 50 °C and speed of 100 rpm. The fungal crude extracts that were obtained after evaporation were weighed and stored in glass vials at 4 °C until use.

### 2.3. Screening of Antibacterial Activity Using In Vitro Assay

#### 2.3.1. Disc Diffusion Assay

The fungal crude extracts were screened for their antibacterial activity against *Aac* JT-0003 using the disc diffusion method described by Bauer et al. [32]. The *Aac* JT-0003 was cultured in NB medium for 48 h and adjusted to contain 5 × 10^8^ CFU/mL based on the 0.5 McFarland Standard. The bacteria were spread on surface of NA medium using a sterile cotton swab.

The dried fungal crude extracts were dissolved in 10% (*v*/*v*) DMSO to a concentration of 60 mg/mL. A sterile paper disc (6 mm in diameter, Whatman no.1) was impregnated with 5 µL of dissolved fungal crude extracts, allowed to dry and placed onto inoculated plates. In each inoculated plate, six paper discs that were soaked in a single fungal crude extract were placed equidistantly. Paper discs impregnated with 10% (*v/v)* DMSO, and cefotaxime (5 µg/disc) were included in each inoculated plate as negative and positive controls, respectively. The inoculated plates were incubated at 28 °C for 24 h and inhibition zones surrounding each disc were measured. The experiments were performed in triplicate for each treatment and the entire experiment was repeated at least twice.

#### 2.3.2. Time Kill Assay

Fungal crude extracts that exhibited antibacterial activity in the disc diffusion assay were further evaluated by using time kill assay as described previously in [33] with some modifications. Fungal crude extracts were prepared at concentrations of 1.5, 3.0 and 6.0 mg. A two-day-old culture of *Aac*, JT-0003 in NB medium was adjusted to 5 × 10^8^ CFU/mL using a spectrophotometer at a wavelength of 600 nm with absorbance value of 0.2. Each of the above-mentioned prepared fungal crude extracts was mixed with 1 mL of bacterial suspension and incubated at 28 °C for 0, 3, and 6 h. Fungal crude extracts without bacterial inoculum were used as negative control, while positive control included the bacterial suspension without fungal crude extracts. Volumes of 10 µL of mixed samples were pipetted and spread over freshly prepared NA medium and plates were incubated at 28 °C. After incubation for 24 h, colonies on individual plates were counted and expressed as the number of colony-forming units/mL (CFU/mL). The percentage of reduction was calculated relatively to the negative control by determining the number of colonies formed as follows;
$$\mathrm{Reduction}\;\mathrm{percentage}\;(\%)=\frac{\mathrm{colony}\;\mathrm{number}\;\mathrm{of}\;\mathrm{control}-\mathrm{colony}\;\mathrm{number}\;\mathrm{of}\;\mathrm{the}\;\mathrm{treatment}}{\mathrm{colony}\;\mathrm{number}\;\mathrm{of}\;\mathrm{control}}\times 100$$

Each treatment was done in triplicate and the entire procedure was repeated twice.

#### 2.3.3. Minimum Inhibitory Concentrations (MICs) by Microdilution Broth

Minimum inhibitory concentrations (MICs) of fungal crude extracts were determined using the microdilution method according to Sarker et al. [34] with some modifications. Briefly, 50 µL of NB medium and 50 µL of normal saline were added into each well of a 96-well microtitration plate. A stock solution of 200 mg/mL was prepared for each fungal crude extract by dissolving the extract in 10% (*v*/*v*) DMSO. Each extract was then prepared with two-fold dilutions to get various final concentrations, and 10 µL of *Aac*, JT-0003 with 5 × 10^8^ CFU/mL was then added to each well. After mixing well, the 96-well microtitration plate was incubated at 28 °C for 24 h. The MIC of the extracts was determined by using a microplate reader (Microplate Software: BioTek Gen5 Data Analysis Software: v2.00.17) at OD 600. The controls included sterile nutrient broth, cefotaxime (1 mg/mL), and 10% (*v*/*v*) DMSO. The MIC value considers lowest extract concentration with no visible growth. The experiment was conducted twice with three replicates per treatment.

### 2.4. Identification of Fungal Endophytes

A combination of morphological and molecular analyses was used to identify four strains of fungal endophytes: MFLUCC 17-0251, MFLUCC 17-0253, MFLUCC 17-0257 and MFLUCC 17-0259. The fungal DNA was extracted from mycelia using the PureDireX Genomic DNA Isolation Kit (Plant, New Taipei City, Taiwan) following the manufacturer’s instructions (BIO-HELIX, Keelung City, Taiwan). The polymerase chain reaction (PCR) was performed using primers that were specific to internal transcribed spacer region (ITS4, 5′-TCCTCCGCTTATTG ATATGC-3′/ITS5, 5′-GGAAGTAAAAGTCGTAACAAGG-3′) [35], translation elongation factor 1-α (EF1-728F, 5′-CATCGAGAAGTTCGAGAAGG-3′/EF1-986, 5′-TACTTGAAGGAACC CTTACC-3′) [36], RNA polymerase II second largest subunit (fRPB2-5F, 5′-GAYGAYMGWGATCAYTTYGG-3′/RPB2-7R, 5′-CCCATWGCYTGCTTMCCCAT-3′) [37], and large subunit ribosomal DNA (LROR, 5′-ACCCGCTGAACTTAAGC-3′/LR5, 5′-TCCTGAGGGAAACTTCG-3′) [38]. PCR was performed under the following conditions: initial denaturation step at 94 °C for 5 min, followed by 37 cycles of denaturation at 94 °C for 30 s, annealing at 55–60 °C for 30 s, and extension at 72 °C for 1.30 min, with a final extension step at 72 °C for 10 min. The amplified DNA fragments were sequenced by SolGent Co., Ltd. (Daejeon, South Korea) using the same primers as for amplification. The sequences of active endophytic fungi were used as queries to perform BLASTn against the nr database to determine contamination and to assemble a dataset. Sequences were aligned using the MAFFT program online (https://mafft.cbrc.jp/alignment/server/, accessed on 16 November 2020). The alignment was trimmed using trimAI software available on the online platform Phylemon (http://phylemon.bioinfo.cipf.es/, accessed on 20 November 2020). Phylogenetic relationships were estimated using maximum likelihood (ML) on CIPRES Science Gateway platform [39]. Sequences of active endophytic fungi were submitted to NCBI GenBank.

### 2.5. Determined Efficiency of Endophytic Fungus on Controlling BFB Infection under Greenhouse Conditions

The endophytic fungi that provided high antagonistic activity against tested pathogens from the three in vitro assays were selected for further analysis at the greenhouse level. Endophytic fungi were grown on V8 medium (200 mL V8, 20 g agar, 3 g CaCO_3_) at 28 °C for 14 days. The conidia were harvested by washing with 5% (*v*/*v*) Tween 20 and conidia characteristics were observed using a compound microscope. The concentration of conidia suspension was adjusted to 1 × 10^6^ conidia/mL by using a hemocytometer. Conidia suspension was mixed into sterile soil at a ratio of 1:9 (spore suspension: soil). Melon and watermelon seeds were sown into the soil mixture and then placed in a greenhouse under the following conditions: 25–28 °C, 60% humidity, and 16 h/8 h of light/dark. Each treatment was arranged in a randomized complete block design (RCBD), and had three replicates of 27 seeds. A two-day-old *Aac*, JT-0003 cultured in NB medium (5 × 10^8^ CFU/mL) was inoculated at the dorsal side of the cotyledon of a two-week-old seedling. Seedlings treated with water and copper hydroxide (1 mg/mL, Funguran, Bangkok, Thailand) were used as negative and positive controls, respectively. The plants were scored using the disease index, when the disease symptoms began to appear. The disease index was scored based on symptoms of infected leaves, as described by [40] with modifications: 0 = no symptoms; 1 = clear soaking lesion and lesion start to spread (irregular shape); 2 = soaking lesion starts to turn brown some part of leave dead; 3 = whole at the lesion, plant cell dies at the area of inoculation; and 4 = death. Disease severity and disease symptoms were calculated as follows:(1)$$\mathrm{Disease}\;\mathrm{severity}\left(\%\right)=\frac{\sum \mathrm{the}\;\mathrm{number}\;\mathrm{of}\;\mathrm{disease}\;\mathrm{leaves}\;\mathrm{in}\;\mathrm{each}\;\mathrm{grade}\times \mathrm{grade}}{\mathrm{total}\;\mathrm{number}\;\mathrm{of}\;\mathrm{leaves}\;\mathrm{investigated}\times \mathrm{the}\;\mathrm{highest}\;\mathrm{disease}\;\mathrm{index}}\times 100$$

The experiment was conducted twice, and disease severity percentages among the treatments were compared using analysis of variance (ANOVA) in SPSS software (SPSS Inc., Chicago, IL, USA, v20) and Duncan’s multiple range test at *p* = 0.05 significance level.

### 2.6. Secondary Metabolites of Neocosmospora sp. MFLUCC 17-0253

Culture filtrate of *Neocosmospora* sp. MFLUCC 17-0253 was extracted using EtOAc. The extract was evaporated under reduced pressure to yield 1.0 of residue. The residue was subjected to Sephadex LH-20, eluted with 100% MeOH to obtain 6 fractions (A‒F). Fraction C (378 mg) was subjected to silica gel 100 eluted with 30% acetone-hexane to give mixture of compounds **1a** and **1b** (2.9 mg). Fraction D (108 mg) was run through silica gel and eluted with 30% acetone-hexane to give four subfractions (D1–D4) and compound 2 (5.2 mg). Subfraction D3 (24.4 mg) was purified by PTLC eluted with 30% acetone-hexane to yield compound 3 (2.4 mg).

### 2.7. Antibiofilm Potential of Neocosmospora sp. MFLUCC 17-0253

To determine the efficacy of *Neocosmospora* sp. MFLUCC 17-0253 crude extract in inhibiting biofilm formation, 96-well microtiter plate method was applied from Shukla and Rao [41] with some modification. Individual wells of the sterile microtiter plate were filled with 180 µL of Mueller Hinton broth (MH broth; BD Difco™, New Jersey, NJ, USA) supplemented with glucose (1% *w*/*v*) and inoculated with 10 µL of overnight *Aac* culture in NB medium (5 × 10^8^ CFU/mL). Ten microliters of fungal crude extract were added from the stock so that the final concentration of fungal crude extract was between 12.5–100 µg/mL. The microtiter plates were incubated for 22 h at 28 °C. After incubation, biofilms were fixed with 99% (*v*/*v*) methanol and washed twice with 200 µL of phosphate buffer saline (PBS, pH 7.2). Plates were dried for 30 min at 60 °C. Then, 200 µL of crystal violet solution (0.2% *w*/*v*) was added to all wells for 5 min. Excess stain was rinsed by thoroughly washing with sterilized distilled water and plates were left to dry. After drying, 150 μL of 33% (*v*/*v*) acetic acid was added to the wells. The absorbance at 570 nm was measured on a multi-detection microplate reader (Synergy HT, Biotek, Winooski, VT, USA). The percentage of biofilm inhibition was calculated using the equation:(2)$$\mathrm{Biofilm}\;\mathrm{inhibition}\;(\%)=1-\frac{\mathrm{OD}570\;\mathrm{of}\;\mathrm{cell}\;\mathrm{treated}\;\mathrm{with}\;\mathrm{fungal}\;\mathrm{crude}\;\mathrm{extract}}{\mathrm{OD}570\;\mathrm{of}\;\mathrm{non}-\mathrm{treated}\;\mathrm{control}}\times 100$$

Experiment was performed in triplicate. The data were then averaged and the standard deviation was calculated. The controls included sterile MH broth, cefotaxime (50 µg/mL), and 10% (*v*/*v*) DMSO.

### 2.8. Assessment of Endophytic Colonization and Determination of Organic Compounds in Extracts Using LC-QTOF-MS

Melon seeds were germinated in two treatments including control (soil mixed with water at a ratio of 1:9), and *Neocosmospora* sp. MFLUCC 17-0253 (soil mixed with endophytic spores at a ratio of 1:9) as described above. Leaf tissues of 14-day-old seedlings were thoroughly washed with running water to remove dust and debris. Leaves were cut into 2 by 2 mm pieces and samples were subjected to surface sterilization with 2% (*v*/*v*) sodium hypochlorite solution for 2 min. After that, the samples were rinsed with 70% (*v*/*v*) ethanol for 20 s and rinsed twice with sterile distilled water. Finally, samples were placed on PDA medium using an aseptic technique. Plates were incubated at 28 °C and inspected every 2–3 days for 3 weeks to observe and record fungal outgrowth [42]. The DNA sequences of *Neocosmospora* sp. MFLUCC 17-0253 at the translation elongation factor 1-α region were compared to sequence of fungal outgrowth obtained from the plant samples by using William Pearson’s lalign program (https://embnet.vital-it.ch/software/LALIGN_form.html, 22 December 2020).

The chemical profiles of 10 in vitro seedling extracts were characterized by using LC-QTOF-MS. Two-week-old seedlings were collected from two treatments as described above (5 seedlings/treatment) and were dried under 50 °C for 2 days or until completely dry. Dried seedlings were then cut into small pieces and soaked in the EtOAc for 2 days. After separating leaf tissue using filtration, the EtOAc was removed by a rotary evaporator under reduced pressure at 50 °C at 100 rpm. Chemical characterization was carried out using the miner modified method in [43] using Agilent 1290 Infinity II UHPLC Systems equipped with an Agilent G6545B QTOF/MS system (Agilent Technologies, Santa Clara, CA, USA). The separation was carried out using an Agilent Poroshell 120 EC-C18 (150 mm × 2.1 mm, 2.7 µm) and a temperature column at 30 °C. Mobile phase A was 0.1 % formic acid in HPLC/UHPLC-water (J.T. Baker Inc, Phillipsburg, NJ, USA), and mobile phase B was 0.1 % formic acid in acetonitrile (ACN). Both mobile phase A and B were degassed at 21 °C for 15 min. The extracted samples were filtered using 0.22 μM-syringe filters and transferred into HPLC vials. The LC-flow rate was set to be 0.2 mL/min and the injection volume was 1 µL for each sample. Gradient elution was performed by a mixture of mobile phase A and B following program: 0–10 min, 5% B; 10–20 min, 17% B; 20–25 min, 100% B. At the end of the program, the eluent composition was back to the initial gradient and the column was equilibrated for 8 min before the next injection.

Agilent dual jet steam electrospray ionization (Dual AJS ESI) was used as a source in operating both negative and positive modes. The mass spectrometry conditions were set as follows: sheath gas flow at 12 L/min, sheath gas temperature at 300 °C, nebulizer gas pressure at 45 psi, gas flow at 11 L/min, and gas temperature at 300 °C. The scan rate for MS and MS/MS modes were 1.00 and 2.00 spectra/s, respectively. Internal reference masses for positive mode were 121.050873 and 922.009798, while for negative mode these were 112.98558700 and 1033.98810900. Mass spectra in the m/z range 50 to 1100 were obtained. For isolation width MS/MS Narrow was ~1.3 amu. Used fixed collision energies were 10.00, 20.00, 40.00 eV. Data acquisition and analysis were performed using Agilent LC-MS-QTOF MassHunter data acquisition software. To classify the compound found in each extract, mass/change, retention time and intensity parameters of the obtained chemical profiles of whole samples were analyzed using univariate and multivariate data analysis, such as principal component analysis (PCA). The data features were filtered based on standard deviation before processing. Normalization of data was performed using the sample median method. Finally, multivariate data analysis of the filtered data was performed with the MetaboAnalyst^®^ online program (https://www.metaboanalyst.ca/, accessed on 22 September 2020).

## 3. Results

### 3.1. Antibacterial Activity of Fungal Crude Extracts

Four antibiotic agents including amoxicillin, streptomycin, ampicillin and cefotaxime were tested for antibacterial activity by disc diffusion assay. Cefotaxime demonstrated the highest antibacterial activity against *Aac*, JT-0003, thus cefotaxime was selected to use as positive control. Out of fifty fungal crude extracts, MFLUCC 17-0251, MFLUCC 17-0253, MFLUCC 17-0257 and MFLUCC 17-0259 exhibited antibacterial activity against *Aac*, with a zone of inhibition ranging from 12.15 mm to 14.58 mm (Table 2). MFLUCC 17-0251 showed the maximum zone of inhibition (14.58 mm) followed by MFLUCC 17-0253 (14.11 mm). Both inhibition zones were larger than that of the antibiotic cefotaxime (13.88 mm). The four fungal crude extracts were then subjected to the microdilution tests to determine the MICs. Similar to the disc diffusion assay, the lowest MIC was 12.5 µg/mL and it was observed in both MFLUCC 17-0251 and MFLUCC 17-0253. Both MICs were lower than the MIC of antibiotic cefotaxime (50 µg/mL; Table 2).

Time kill assay was used to further analyze for colony forming efficiency of four fungal crude extracts that exhibited high antibacterial activity from previous assays. Since the highest MIC among the four fungal crude extracts was 200 µg/mL (MFLUCC 17-0257), time kill assays were performed with 150 to 600 µg/mL fungal crude extracts. The maximum reduction percentage was observed at 6-h-long treatments. The MFLUCC 17-0251 (0.6 mg) and MFLUCC 17-0253 (0.6 mg) showed the highest reduction of *Aac*, JT-0003 at 97.51% and 96.25%, respectively (Figure 1 and Figure 2). These data indicated that only 0.6 mg of active compounds from those two strains were required to kill the cells. Interestingly, three fungal crude extracts required a minimum of 3 h to inhibit colony formation of *Aac*, JT-0003 except for MFLUCC 17-0251 that showed up to 60% of reduction at 0 h of incubation indicating that its crude extract had higher efficiency in inhibiting colony formation compared to the other crude extracts (Figure 1 and Figure 2).

The data from all three assays indicated significant antibacterial activity of four endophytic fungi and those were identified by phylogenetic tree analysis and morphological characteristic examinations. The phylogenetic tree analysis revealed that all four fungal endophytes belonged to *Neocosmospora* (Figure 3). Their colonies on PDA covered the entire plate after 2 weeks at 28 °C (Figure 4). Their mycelia were velvety and moderately fluffy with an irregular margin; *Neocosmospora* sp. MFLUCC 17-0251 was initially light-orange, and later became white; *Neocosmospora* sp. MFLUCC 17-0253’s surface was initially orange and later became white; *Neocosmospora* sp. MFLUCC 17-0257 had a white colony and then became off-white; *Neocosmospora* sp. MFLUCC 17-0259’s colony was orange brown in the center and white at the margin.

### 3.2. Effect of Antagonistic Endophytic Fungi on BFB Infection in the Greenhouse

The four fungal strains were evaluated for their ability to control *Aac* in watermelon and melon seedlings. Seedlings from all treatments were successfully germinated (98–100 germination percentage) with no significant differences found among treatments. The watermelon and melon cotyledons in the untreated (inoculated) control had severe symptoms within two to four day after inoculation (Figure 5 and Figure 6A). The watermelon and melon cotyledons in the negative control (without inoculation of *Aac*) were healthy (Figure 6B).

Even though *Neocosmospora* sp. MFLUCC 17-0251 and MFLUCC 17-0253 consistently provided extensive antibacterial activity for all in vitro assays, only the latter showed strong suppression of bacterial fruit blotch symptoms in watermelon seedlings. The disease severity of the watermelon cotyledon in *Neocosmospora* sp. MFLUCC 17-0253 treatment was 0.00–10.46%, compared to the 7.73–46.71% of the untreated (inoculated) control treatment (Figure 6A). MFLUCC 17-0257 and MFLUCC 17-0259 did not differ significantly (*p* < 0.05) from the untreated inoculated control. Soil treated with all four fungal endophytic strains, and copper hydroxide had intermediate efficacy in suppressing bacterial fruit blotch symptoms.

Similar results were found in melon seedlings. The treatment of *Neocosmospora* sp. MFLUCC 17-0253 significantly (*p* < 0.05) decreased disease severity of BFB, and showed the lowest disease severity (0.00–12.04%), when compared with the untreated control (10.65–53.76%). MFLUCC 17-0257 and MFLUCC 17-0259 treatments did not significantly differ (*p* < 0.05) from the untreated controls. Soil treated with all endophytic fungal strains, and *Neocosmospora* sp. MFLUCC 17-0251 had intermediate efficacy in suppressing bacterial fruit blotch symptoms.

### 3.3. Secondary Metabolites of Neocosmospora sp. MFLUCC 17-0253

The secondary metabolites were identified as 2-methoxy-6-methyl-7-acetonyl-8-hydroxy-1,4-maphthalenedione (**1a**), 5,8-dihydroxy-7-acetonyl-1,4-naphthalenedione (**1b**) [44], anhydrojavanicin (**2**) [44], and fusarnaphthoquinones B (**3**) [45] shown in Figure 7. The structures of these metabolites were identified by comparing them with NMR data available in the literature (Appendix A). The isolated compounds **1**–**3** were evaluated for their antibacterial activity against *Aac*. All three compounds showed potent antibacterial activity with an MIC value of 0.0075 mg/mL (mixture of 2-methoxy-6-methyl-7-acetonyl-8-hydroxy-1,4-maphthalenedione and 5,8-dihydroxy-7-acetonyl-1,4-naphthalenedione), 0.004 mg/mL (anhydrojavanicin), 0.025 mg/mL (fusarnaphthoquinones B). Importantly, the combination of these three compounds showed antibacterial activity with an MIC value of 0.002 mg/mL.

### 3.4. Antibiofilm Potential of Neocosmospora sp. MFLUCC 17-0253

The major compounds present in *Neocosmospora* sp. MFLUCC 17-0253 were identified as naphthoquinones and its derivatives as described above. Previous studies have reported that nearly 80% of all microbial infections by this group of compounds were found to be associated with biofilm formation [46]. In this work, efforts have been made to investigate, whether the tested concentrations of fungal crude extract of *Neocosmospora* sp. MFLUCC 17-0253 could attenuate the development of biofilm. The result showed that there was a significant difference in biofilm formation between untreated (negative control) and treated with fungal crude extract *Neocosmospora* sp. MFLUCC 17-0253. The treatment of negative control and treat with 10% DMSO did not show any significant decrease in biofilm formation. The amount of biofilm formation sharply decreased as the concentration of fungal crude extract increased. Treatment with fungal crude extract of *Neocosmospora* sp. MFLUCC 17-0253 concentration of 12.5–100 µg/mL resulted in a significant decrease of 44–77% (Figure 8). More importantly, the result demonstrated that *Neocosmospora* sp. MFLUCC 17-0253 highly inhibited biofilm formation (73% inhibition) compared to cefotaxime (49% inhibition) at the same concentration.

### 3.5. Determination of Organic Compounds in Extracts Using LC-QTOF-MS

Metabolic profiles of the extracted samples were obtained using LC-Q/TOF-MS in the positive and negative ion modes. The PCA score plots in the control and treatment groups showed clear separation in both negative (Figure 9a) and positive ion modes (Figure 9b). Univariate analysis, such as volcano plots with fold change threshold (x) 3 and *t*-tests threshold (y) 0.05 in the positive and negative ion modes were also performed (Figure 10). Difference of compounds between the studied groups was noted. The pink circles represent chemicals above the threshold plotted as log scales. In the assessment of endophytic colonization, we were able to confirm that fungal outgrowth from the plant samples that were inoculated with *Neocosmospora* sp. MFLUCC 17-0253 had similar DNA sequence (94.3%) to that of inoculated endophytes (data not shown). These data suggested that *Neocosmospora* sp. MFLUCC 17-0253 successfully colonized the host plant (melon) and was responsible for changing the metabolic profiles of those plants.

## 4. Discussion

Bacterial fruit blotch (BFB) caused by *A. citrulli* is a serious seed borne disease with a destructive impact on cucurbit crops. Several strategies have been employed along the production chain to protect against the disease and to avoid major crop losses. Seed producers, transplant growers, and commercial growers have all participated in these efforts. However, all these strategies have limitations. For instance, oxalic acid and culture filtrates of *Aspergillus niger* Y-1 have been used to suppress *Aac* in watermelon seedlings, but this negatively affected seed germination of watermelon [7]. Johnson et al. (2011) [47] used nonpathogenic *A. citrulli* on female watermelon blossoms and watermelon seeds as a means to control the disease. The fungus showed potential as a biological control for seed treatment; unfortunately, this method has not yet been tested commercially. Similarly, treatments with high concentration of peroxyacetic acid, dry heat treatment, chlorine gas exposure, and acidic electrolyzed water had adverse effects on raising the germination percentage and quality of treated seed [8,48,49]. Thus, our study investigated bioactive compounds obtained from endophytic fungi in an effort to find alternative strategies to control *A. citrull**i.* Our results showed that the fungal endophyte *Neocosmospora* sp. MFLUCC 17-0253 has high antagonistic activity against *Aac*, JT-0003. More importantly, this fungal endophyte did not have a negative effect on seed germination or growth of either melon or watermelon, as shown by the similar germination percentages between seedlings treated with spores of fungal endophytes and untreated soil.

Throughout this study, several genera of endophytic fungi including *Nemania* sp., *Pestalotiopsis* sp. *Hypoxylon griseobrunneum*, *Diaporthe* sp., *Xylaria* sp. and *Colletotrichum* sp., all of which were isolated from mangrove forests and tea plantations were tested for their ability to control bacterial fruit blotch disease. In vitro bioassays showed excellent suppressive effects for four strains of fungal endophytes (MFLUCC 17-0251, MFLUCC 17-0253, MFLUCC 17-0257 and MFLUCC 17-0259), all of which belonged to the genus *Neocosmospora* (Hypocreales, Nectriaceae). The genus is widely distributed and comprises fungi commonly found in soil, plant debris, living plant material, air and water [50]. *Neocosmospora* is also known as producer of bioactive natural products including antibacterial agents, cytotoxic compounds like the immunosuppressive agent’s cyclosporine A and C, and naphthoquinones [51,52,53]. Members of the genus produce a wide range of toxins displaying activities against plants and animals, cell cultures and diverse microorganisms, thus *Neocosmospora* has also been sporadically associated with human and animal mycotoxicosis [54,55,56]. The list of known toxic metabolites includes furanoterpenoids, ipomeanols and ipomeanine [55,57], naphthazarins [58], while the alleged production of the trichothecenes scirpentriol, NT-2, T-1, T-2 toxins and neosolaniol are most likely based on misidentified isolates [54,59,60].

The 2-methoxy-6-methyl-7-acetonyl-8-hydroxy-1,4-maphthalenedione, 5,8-dihydroxy-7-acetonyl-1,4-naphthalenedione, anhydrojavanicin and fusarnaphthoquinones B were responsible for inhibiting *Aac*, JT-0003. These compounds belong to the group of naphthoquinones, secondary metabolites produced by plants, actinomycetes, fungi, lichens and algae. Naphthoquinones display various types of biological active profiles including antibacterial, anti-diabetic, antifungal, anticancer, and anti-inflammatory activities [61,62,63,64]. For instance, the effects of Juglone on *Staphylococcus aureus*, lawsone on *Escherichia coli* and plumbagin on *Candida albicans* were associated with oxidative stress and/or disrupted thiol metabolism. The literature contains several reports concerning synthetic derivatives that are effective against *Mycobacterium tuberculosis* such as amino-1,4-naphthoquinone-appended triazoles [65] and plumbagin (5-hydroxy-2-methyl-1, 4-naphthoquinone), a bicyclic naphthoquinone from *Plumbago* species. These derivatives are active in combination with oxacillin and tetracycline against methicillin-resistant *Staphylococcus aureus* (MRSA) [66]. The antimicrobial activity of plumbagin is due to its ability to chelate several trace metals, inhibit cytokinesis and induce ROS production.

Fungal naphthoquinones have many physiological roles especially antifungal and antibacterial activity. The ability of this metabolite to change the sensitivity of its target and interact with oxidative system of the cells and modify metabolism is responsible for its observed toxicity to fungi [67]. On the other hand, inhibition of mitochondrial respiration, suppression of RNA biosynthesis, and ability of redox conversions, cytotoxic, and phytotoxic properties due to interactions with the oxidative systems of cells provide its antibacterial activity. The ability to synthesize naphthoquinone is prevalent in *Fusarium* and phylogenetically close species [19,68]. Interestingly, *Neocosmospora* was previously assigned to the *F. solani* species complex (FSSC) [50]. However, secondary metabolites from *Neocosmospora* sp. MFLUCC 17-0253 produced in vitro showed antagonistic activity against the Gram-negative bacterium *A. citrull**i.* This is in contrast to previous studies reporting antibiotic activity, mostly against Gram-positive bacteria [64,69].

To better understand the resistance mechanism of these compounds, we investigated the antibiofilm activity of fungal crude extract *Neocosmospora* sp. MFLUCC 17-0253 against *Aac*. The biofilm formation of *Aac* was inhibited after treatment with the fungal crude extract. The inhibition might result from cellular accumulation of *Aac* reactive oxygen species (ROS). ROS refers to the reactive chemical species containing oxygen like superoxide, peroxide, and singlet oxygen [70]. ROS production can irreversibly damage the cell membrane, DNA, mitochondria and may also induce cell death. Thus, the cytotoxic effect of naphthoquinone compounds is based on generation of ROS and subsequent apoptosis induction [68]. Inhibition of formation of biofilm due to microbial accumulation of ROS has been previously documented [71]. Various derivatives of 1,4-naphthoquinone have been found to inhibit biosynthesis of staphyloxanthin, a virulence factor associated with biofilm formation, thereby making the microorganism susceptible to ROS [72]. Thus, antibiofilm formation activity could be one important mechanism for the observed antibacterial activity of this fungal crude extract. Given that nearly 80% of all microbial infections have been found to be associated with biofilm formation [46].

To the best of our knowledge, this is the first study to report secondary metabolites with significant antibacterial activity against a bacterial disease, for which to date there is still no chemical or other means to control. Importantly, the level of resistance of the endophyte treatment was consistently high not only in the laboratory setting, but also during greenhouse evaluation. Moving forward, future studies should focus on open field experiments and addressing the challenges involved. These include finding suitable fields with low load of pesticides, performing risk assessment and evaluating interrelationships with environmental variables. However, high antibacterial activity that derives from natural resources (in this case fungal endophyte) equipped with employing practical applications could be an alternative method and become an important element in disease management of BFB which will greatly impact the global cucurbit industry.

## 5. Conclusions

In summary, we screened 50 endophytic fungi, which had the possibility of effectively controlling the *Aac*. We found that the endophytic fungus MFLUCC 17-0253 was highly effective in controlling *Aac* both in vitro and in planta. We identified the endophytic fungi MFLUCC 17-0253 as *Neocosmospora*. Naphthoquinones derivatives of potent antibacterial activity were isolated from this strain. The combination of these compounds resulted in stronger antibacterial activity and had an MIC value of 0.002 mg/mL. The chemical profiles of control and treatment groups showed differences in planta. This was the first report of *Neocosmospora* sp. MFLUCC 17-0253 showing that it has the potential to be used as a biocontrol agent to prevent BFB.

## Figures and Tables

**Figure 1 jof-07-00370-f001:**
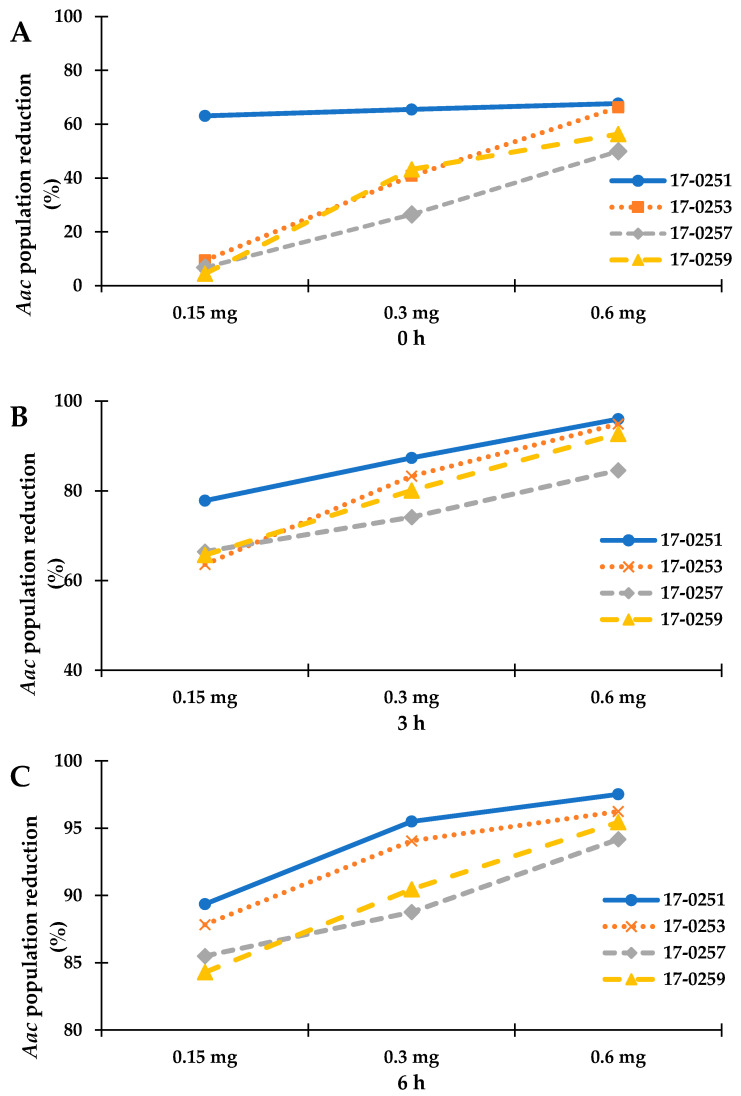
Represent the time kill assay graph of endophytic fungal crude extracts against *Aac*, JT-0003; (**A**) incubated for 0 h, (**B**) incubated for 3 h, and (**C**) incubated for 6 h. Exponential graph of time of *Neocosmospora* sp. MFLUCC 17-0251, MFLUCC 17-0253, MFLUCC 17-0257 and MFLUCC 17-0259 were diluted in 10% DMSO and tested at the indicated concentrations.

**Figure 2 jof-07-00370-f002:**
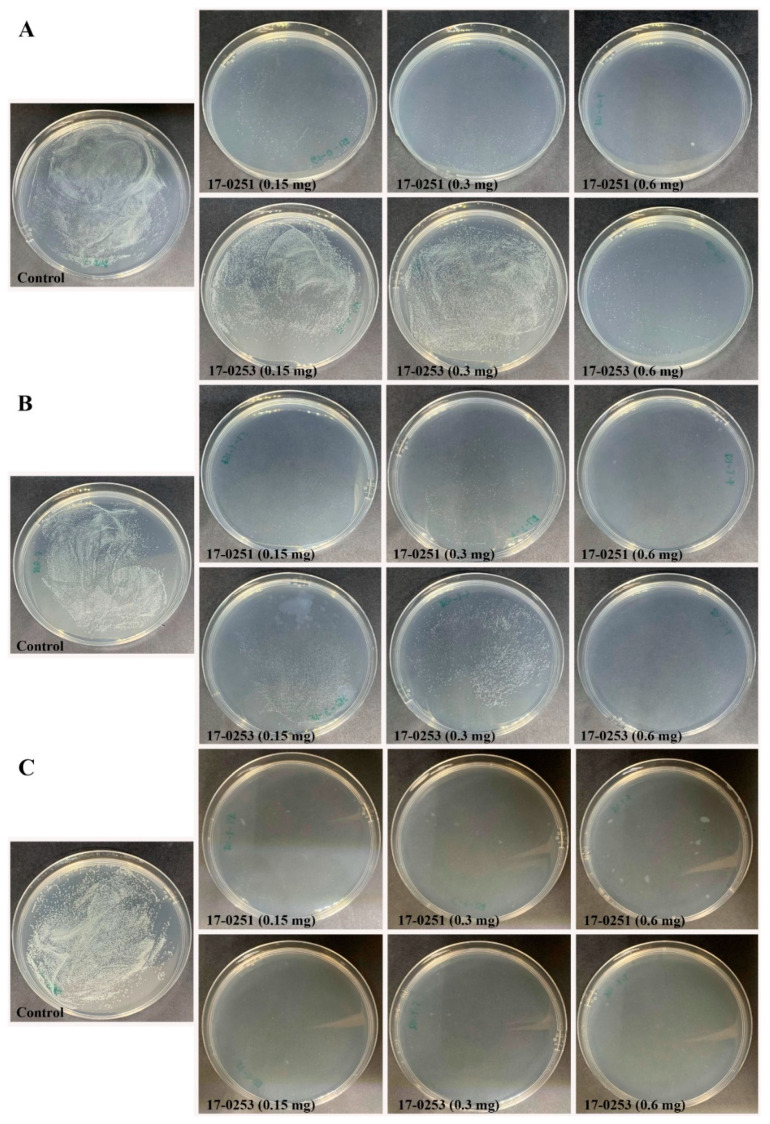
Efficacy of the time kill assay of endophytic fungi crude extract *Neocosmospora* sp. MFLUCC 17-0251 and MFLUCC 17-0253 in suppression of *Aac*, JT-0003 after 24 h. Inhibition of *Aac* growth at (**A**) 0 h, (**B**) 3 h, and (**C**) 6 h. Control refers to bacterial suspension cultured without fungal crude extracts.

**Figure 3 jof-07-00370-f003:**
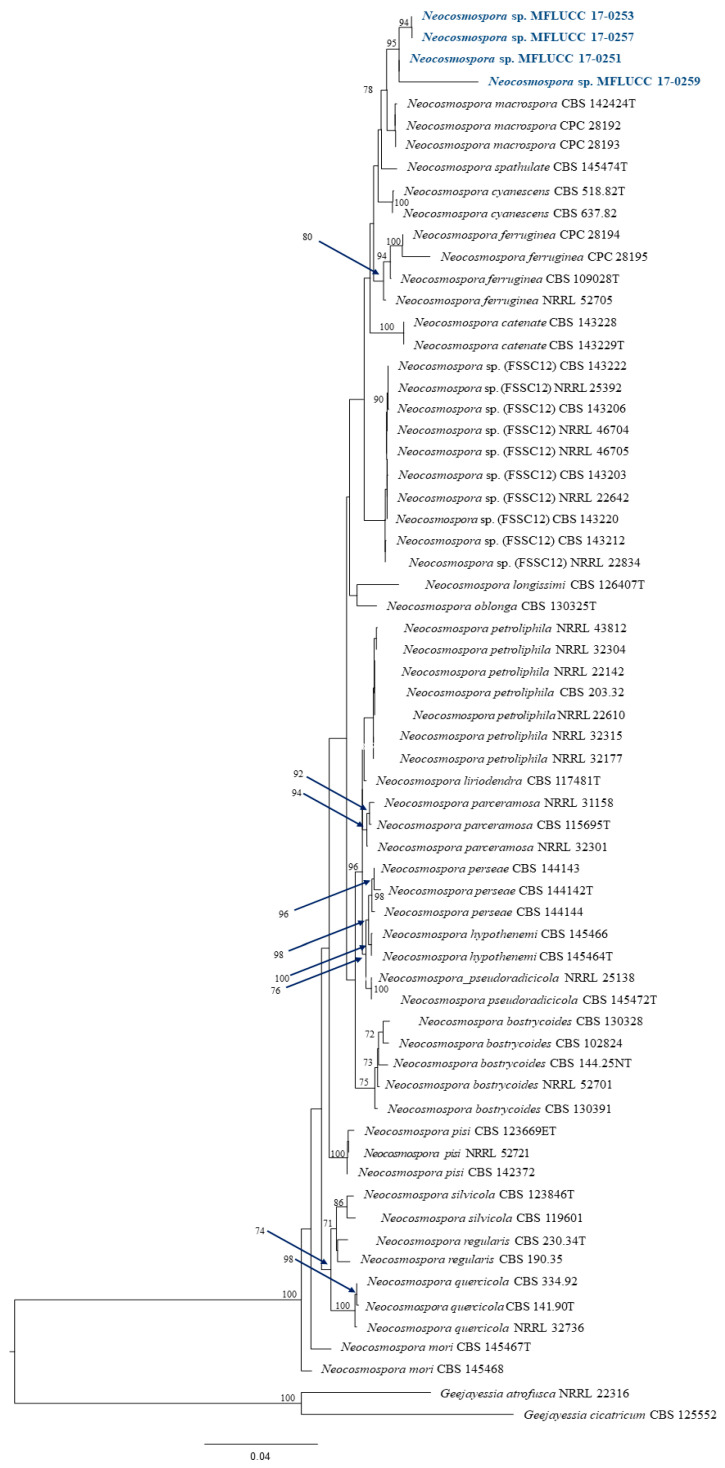
Phylogram generated from a RAxML analysis based on ITS, TEF1, LSU and RPB2 sequence data. Bootstrap support values for ML equal to or greater than 60% above the nodes. The tree is rooted to *Geejayessia atrofusca* (NRRL 22316) and *Geejayessia cicatricum* (CBS 125552), Blue arrows indicate bootstrap support for the node at which they point.

**Figure 4 jof-07-00370-f004:**
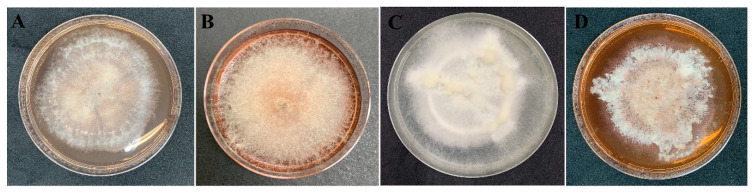
Morphology characteristic of endophytic fungi on PDA medium after 2 weeks. *Neocosmospora* sp. (**A**) MFLUCC 17-0251, (**B**) MFLUCC 17-0253, (**C**) MFLUCC 17-0257, and (**D**) MFLUCC 17-0259.

**Figure 5 jof-07-00370-f005:**
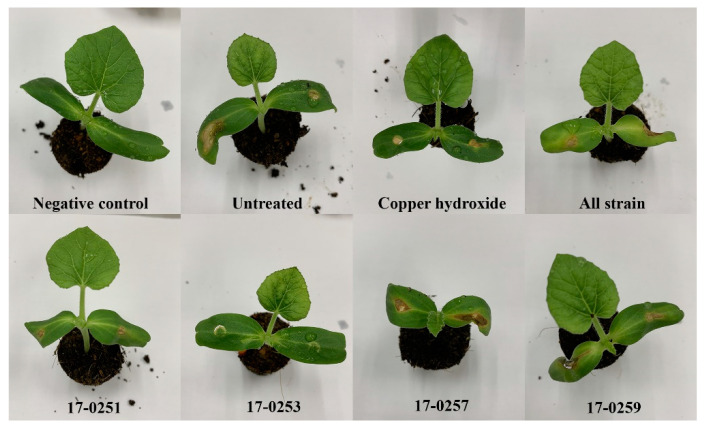
Effect of soil treatment of leaves of melon at seedling stage with cultures of endophytic fungi and copper hydroxide caused by *Aac*, JT-0003.

**Figure 6 jof-07-00370-f006:**
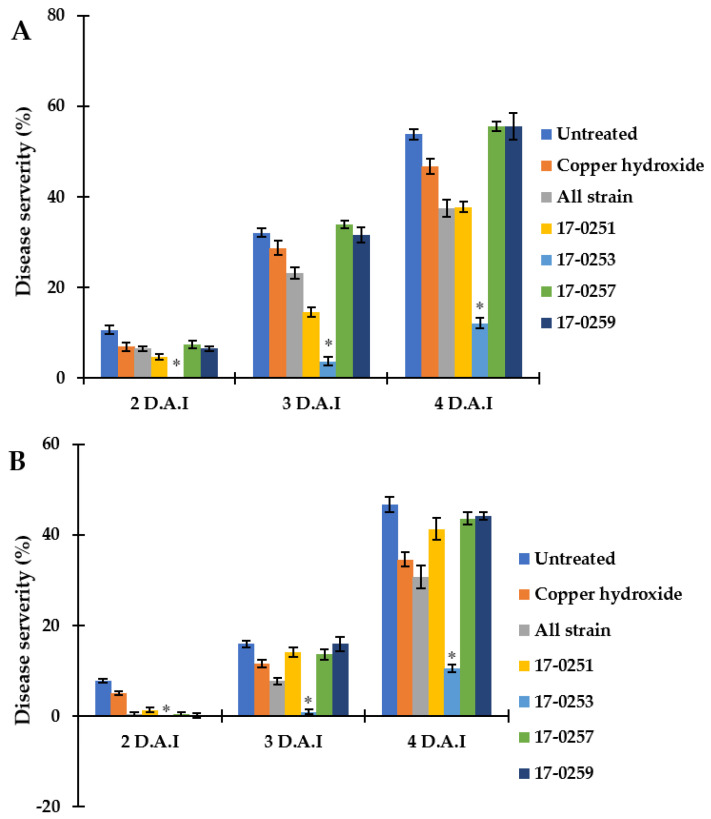
Efficacy of soil treatments with culture of endophytic fungi and copper hydroxide at seedling stage of watermelon and melon caused by *Aac*, JT-0003 under greenhouse. A histogram showing difference in disease severity percentage of *Aac* of (**A**) watermelon and (**B**) melon for seven treatments. Values are the means of two experiments, each treatment with three replicates. Vertical bars represent the standard error of means (n = 2). Asterisks represent treatments in which disease severity was significantly (* *p* < 0.05) reduced compared to control treatment.

**Figure 7 jof-07-00370-f007:**
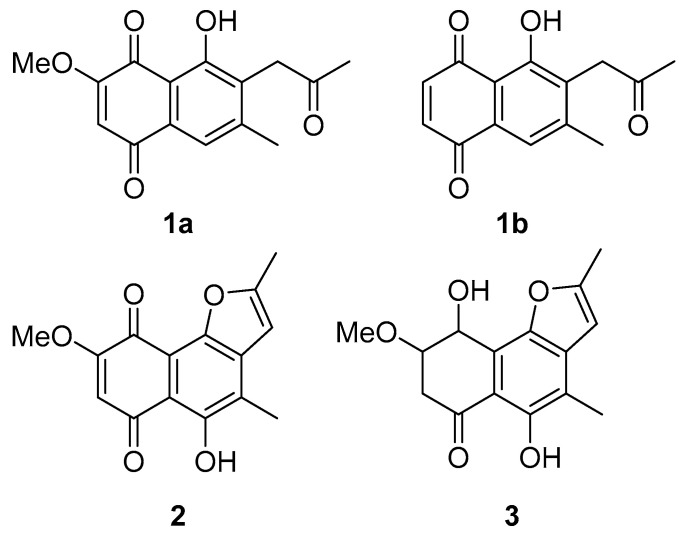
Structures of isolated compounds from *Neocosmospora* sp. MFLUCC 17-0253. (**1a**) Mixture of 2-methoxy-6-methyl-7-acetonyl-8-hydroxy-1,4-maphthalenedione and (**1b**) 5,8-dihydroxy-7-acetonyl-1,4-naphthalenedione, (**2**) anhydrojavanicin, and (**3**) fusarnaphthoquinones B.

**Figure 8 jof-07-00370-f008:**
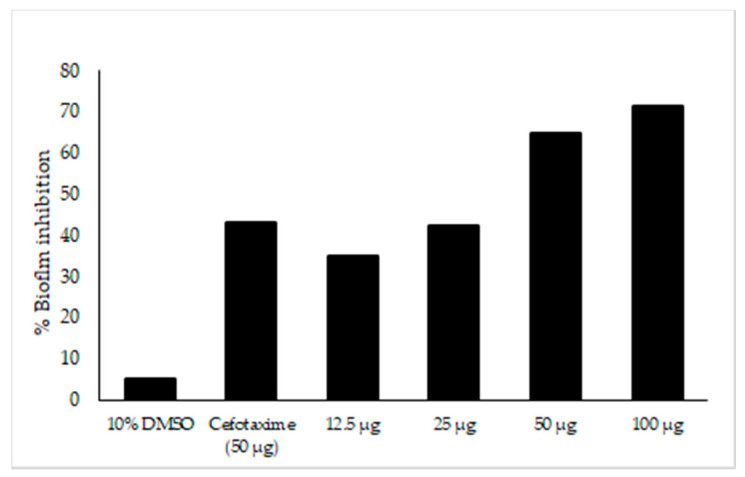
Effect of fungal crude extract *Neocosmospora* sp. MFLUCC 17-0253 on biofilm inhibition quantified by crystal violet. Biofilm formed in 96-well microtiter plate for 22 h was treated with different concentrations of fungal crude extract. The data represented by the mean of the results of a total of twelve readings.

**Figure 9 jof-07-00370-f009:**
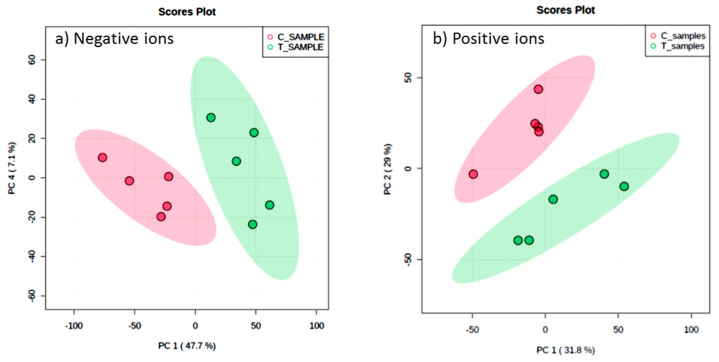
Scores plot between the selected principal components (PCs) in both ion modes of extracted samples; negative ion mode (**a**) and positive ion mode (**b**) groups, the control (C) and treatment (T) groups.

**Figure 10 jof-07-00370-f010:**
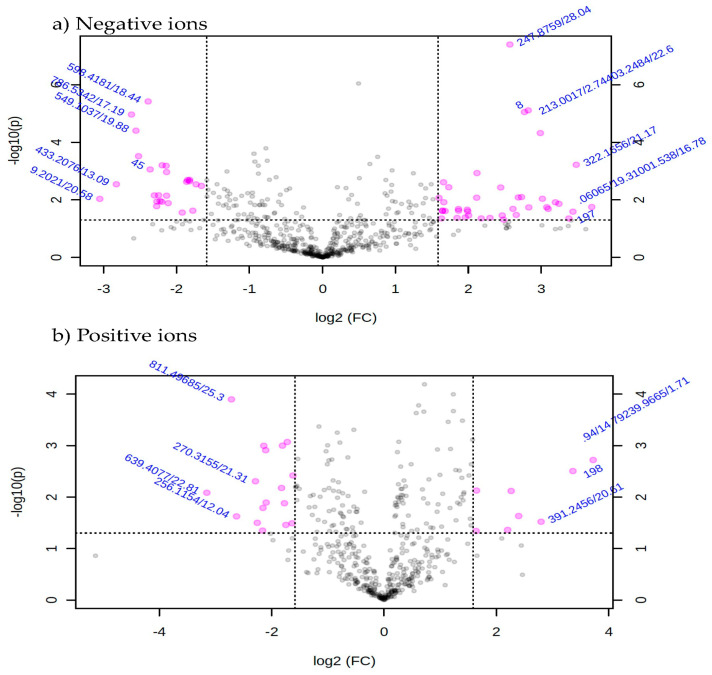
Volcano plot with fold change threshold (x) 3 and *t*-tests threshold (y) 0.05 principal components (PCs) in both ion modes of extracted samples; negative ion mode (**a**) and positive ion mode (**b**) groups. The pink circles represent features above the threshold.

**Table 1 jof-07-00370-t001:** List of fungal endophytes that used in this study.

MFLUCC Code	*Fungal* sp.	Plant Source	Host	References
17-0005	*Nemania* sp.	Leaf	*Rhizophora apiculata*	[31]
17-0011	N/A	Petiole	*R. mucronata*	N/A
17-0012	N/A	Petiole	*R. mucronata*	N/A
17-0014	N/A	Petiole	*R. mucronata*	N/A
17-0018	*Pestalotiopsis* sp.	Aerial stilt root	*R. apiculata*	[31]
17-0024	*Pestalotiopsis* sp.	Aerial stilt root	*R. apiculata*	[31]
17-0027	*Hypoxylon griseobrunneum*	Leaf	*R. apiculata*	[31]
17-0034	N/A	Aerial stilt root	*R. apiculata*	N/A
17-0251	*Neocosmospora* sp.	N/A	*Xylocarpus granarum*	N/A
17-0253	*Neocosmospora* sp.	N/A	*R. apiculata*	N/A
17-0257	*Neocosmospora* sp.	N/A	*R. mucronata*	N/A
17-0259	*Neocosmospora* sp.	N/A	*Ceriopsis decandra*	N/A
17-0416	N/A	N/A	*R. apiculata*	N/A
17-0421	N/A	N/A	*R. mucronata*	N/A
17-0447	N/A	N/A	*Avicennia marina*	N/A
17-0522	N/A	N/A	*R. mucronata*	N/A
17-0530	N/A	N/A	*C. decandra*	N/A
T19-1054	*Nigrospora* sp.	Petiole	*Camellia sinensis*	N/A
T19-1055	N/A	Petiole	*C. sinensis*	N/A
T19-1056	*Fusarium* sp.	Petiole	*C. sinensis*	N/A
T19-1057	N/A	Petiole	*C. sinensis*	N/A
T19-1058	*Pestalotiopsis* sp.	Petiole	*C. sinensis*	N/A
T19-1062	*Neopestalotiopsis* sp.	Petiole	*C. sinensis*	N/A
T19-1063	*Neopestalotiopsis* sp.	Petiole	*C. sinensis*	N/A
T19-1064	*Phormopsis* sp.	Leaf	*C. sinensis*	N/A
T19-1065	N/A	Leaf	*C. sinensis*	N/A
T19-1066	*Diaporthe* sp.	Leaf	*C. sinensis*	N/A
T19-1067	N/A	Leaf	*C. sinensis*	N/A
T19-1068	*Nigrospora* sp.	Leaf	*C. sinensis*	N/A
T19-1071	*Collectotrichum* sp.	Leaf	*C. sinensis*	N/A
T19-1072	N/A	Leaf	*C. sinensis*	N/A
T19-1077	N/A	Leaf	*C. sinensis*	N/A
T19-1078	*Pseudopeatalotiopsis* sp.	Leaf	*C. sinensis*	N/A
T19-1079	N/A	Leaf	*C. sinensis*	N/A
T19-1081	*Pseudopeatalotiopsis* sp.	Root	*C. sinensis*	N/A
T19-1082	*Fusarium* sp.	Root	*C. sinensis*	N/A
T19-1083	*Fusarium* sp.	Root	*C. sinensis*	N/A
T19-1084	N/A	Root	*C. sinensis*	N/A
T19-1086	*Fusarium* sp.	Root	*C. sinensis*	N/A
T19-1336	*Pestalotiopsis* sp.	Petiole	*C. sinensis*	N/A
T19-1339	*Neopestalotiopsis* sp.	Petiole	*C. sinensis*	N/A
T19-1340	*Clonostachys* sp.	Petiole	*C. sinensis*	N/A
T19-1342	*Phormopsis* sp.	Petiole	*C. sinensis*	N/A
T19-1343	N/A	Petiole	*C. sinensis*	N/A
T19-1344	N/A	Leaf	*C. sinensis*	N/A
T19-1346	*Neopestalotiopsis* sp.	Leaf	*C. sinensis*	N/A
T19-1348	N/A	Leaf	*C. sinensis*	N/A
T19-1349	*Clonostachys* sp.	Leaf	*C. sinensis*	N/A
T19-1351	*Pesudopeatalotiopsis* sp.	Leaf	*C. sinensis*	N/A
T19-1352	N/A	Leaf	*C. sinensis*	N/A
T19-1353	*Neopestalotiopsis* sp.	Leaf	*C. sinensis*	N/A
T19-1354	N/A	Leaf	*C. sinensis*	N/A
T19-1355	N/A	Leaf	*C. sinensis*	N/A
T19-1356	*Fusarium* sp.	Leaf	*C. sinensis*	N/A
T19-1359	*Fusarium* sp.	Leaf	*C. sinensis*	N/A
T19-1360	N/A	Leaf	*C. sinensis*	N/A
T19-1362	*Diaporthe* sp.	Leaf	*C. sinensis*	N/A
T19-1363	N/A	Leaf	*C. sinensis*	N/A
T19-1365	*Hypoxylon*	Root	*C. sinensis*	N/A
T19-1367	*Chaetomium* sp.	Root	*C. sinensis*	N/A
T19-1368	*Diaporthe* sp.	Root	*C. sinensis*	N/A
T19-1369	*Diaporthe* sp.	Root	*C. sinensis*	N/A
T19-1370	*Chaetomium* sp.	Root	*C. sinensis*	N/A
T19-1372	*Chaetomium* sp.	Root	*C. sinensis*	N/A
T19-1373	N/A	Root	*C. sinensis*	N/A

**Table 2 jof-07-00370-t002:** In vitro antibacterial activity of the fungal crude extracts against *Aac*, JT-0003 in disc diffusion assay and MICs by microdilution broth.

Treatment	Zone of Inhibition (mm) ^1^	Minimum Inhibitory Concentrations (µg/mL)
DMSO	0	0
Cefotaxime	13.88 ± 0.67	50.0
17-0251	14.58 ± 0.34	12.5
17-0253	14.11 ± 0.27	12.5
17-0257	12.15 ± 0.78	200
17-0259	13.91 ± 0.78	50.0

^1^ Data are presented as mean ± S.D. values of three independent experiments.

## Data Availability

Not applicable.

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
