# Peer review of "Napthoquinones from Neocosmospora sp.—Antibiotic Activity against Acidovorax citrulli, the Causative Agent of Bacterial Fruit Blotch in Watermelon and Melon"

_jof, 2021, doi:10.3390/jof7050370_

Round 1
Reviewer 1 Report
Brief Summary
The manuscript jof-1182529 reports results on the evaluation of the antimicrobial activities of fungal endophytes isolates, obtained from mangrove forests and tea plantations against Acidovorax citrulli. Antimicrobial activity was studied both in vitro and in planta, investigating also secondary metabolites from the most promising fungal endophytes.
Broad comments
- Introduction: The introduction correctly places the study in a broad context and highlight why it is important. The authors clearly defined the purpose of the work and its significance. The specific hypotheses being tested should be added.
- Materials and Methods: Except for small missing details, the authors described with sufficient details the methods used. See minor comments.
- Results: The results description is clear and precise. See minor comments for some little changes.
- Discussion: Authors correctly discussed the results from the perspective of previous studies and of the working hypotheses in the broadest context possible. I would provide more details about future studies, underling the challenges in the open field experiment tests and the large scale production of the secondary metabolites.
- Conclusions: The section is appropriate.
Specific comments
L13 – Revise Cucurbitaceae italic.
L23 – I would use the term in planta.
L24 – Add “seedlings” after “untreated”.
L85 – Add specific working hypotheses.
L93 – Add UFC/mL within brackets.
L104 – For “Ethyl acetate”, define the abbreviation if used next in the text and be consistent with its use.
L112, L142 – Add the name of the first author or revise the sentence.
L129 – Add details about the wavelength used and absorbance reached for the sample.
L165 – Add details about PCR programs.
L215 – If performed, add details about adjustments or reading made on the overnight culture before use.
L233, L234 – Add ratios of soil/water mixtures.
L241-243 – These lines should be revised for clarity.
L245 – “in vitro seedling” instead of “plant”.
Figure 2 – add a description for controls.
Figure 6 – Revise the y-axis for panel B. Explain the meaning of the asterisk symbol, reporting statistic carried out.
Figure 9 and Figure 10 – Define “PCs”.
L504 –Provide more details about future studies, underling the challenges in the open field experiment tests and the large scale production of the secondary metabolites.
L516 – Remove the hyphen in the term in-vitro.
L516 and L520 – Again, I would use the term in planta.
Some words are not formatted properly (eg “plantations” in L81 or “parameters” in L273).
Reviewer 2 Report
The manuscript "Napthoquinones from Neocosmospora sp. – antibiotic activity against Acidovorax citrulli, the causative agent of bacterial fruit blotch in watermelon and melon" provide significant results and a complete point of view for the research and application of potential biocontrol agent for BFB disease.
Author Response
There is no specific comment to edit or answer